# Social Processes: Self-Supervised Forecasting of Nonverbal Cues in Social Conversations

## Abstract

The default paradigm for the forecasting of human behavior in social conversations is characterized by top-down approaches. These involve identifying predictive relationships between low level nonverbal cues and future semantic events of interest (e.g. turn changes, group leaving). A common hurdle however, is the limited availability of labeled data for supervised learning. In this work, we take the first step in the direction of a bottom-up self-supervised approach in the domain. We formulate the task of Social Cue Forecasting to leverage the larger amount of unlabeled low-level behavior cues, and characterize the modeling challenges involved. To address these, we take a meta-learning approach and propose the Social Process (SP) models—socially aware sequence-to-sequence (Seq2Seq) models within the Neural Process (NP) family. SP models learn extractable representations of non-semantic future cues for each participant, while capturing global uncertainty by jointly reasoning about the future for all members of the group. Evaluation on synthesized and real-world behavior data shows that our SP models achieve higher log-likelihood than the NP baselines, and also highlights important considerations for applying such techniques within the domain of social human interactions.

## 1 Introduction

Picture a situated interactive agent such as a social robot conversing with a group of people. How can agents act in such a setting? We sustain conversations spatially and temporally through explicit behavioral cues—examples include locations of partners, their orientation, gestures, gaze, and floor control actions [1–3]. Evidence suggests that we employ an anticipation of these and other cues to navigate daily social interactions [1, 4–8]. Consequently, the ability to forecast the future constitutes a natural objective towards the realization of machines with social skills. As such, interactive agents typically contend with uncertainties in inferences surrounding cues [3]. So beyond making real-time inferences, such systems may achieve more fluid interactions by leveraging the ability to forecast future states of the conversation [9].

In addition to the development of social agents, behavior forecasting is also of significance in social psychology, where the focus is on gaining insight into human behavior. Since human-interpretability is of essence, top-down approaches largely constitute the default paradigm, where specific events of semantic interest are selected first for consideration and their relationship to potentially predictive cues are studied in isolation—either in controlled interactions in lab settings, or in subsequent statistical analyses [10, 11]. Examples of such semantic events include speaker turn transitions [5, 12, 13], mimicry episodes [14], or the termination of an interaction [9, 15]. However, one hurdle in the top-down paradigm is limited data. The events (that constitute the labels or the dependent variables) often occur infrequently over a longer interaction, reducing the effective amount of labeled data. This precludes the use of neural supervised learning techniques that tend to be data intensive.

Submitted to 35th Conference on Neural Information Processing Systems (NeurIPS 2021). Do not distribute.

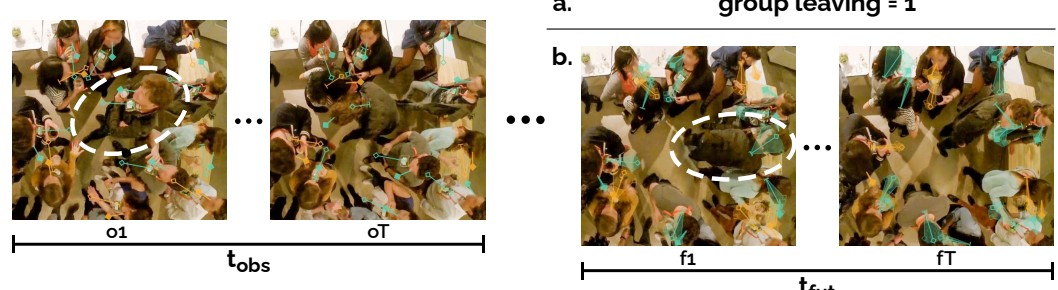

**Figure 1:** Conceptual illustration of forecasting approaches on an in-the-wild conversation from the Match-NMingle dataset [16]. **a.** The top-down approach entails predicting a semantic event or action of interest for the observed window $t_{\mathrm{obs}} \coloneqq [o1 \ldots oT]$. Here we illustrate *group leaving* [15]; the circled individual in the center leaves a group in the future. **b.** In contrast, we propose a bottom-up approach in the social conversation forecasting domain through the task of *Social Cue Forecasting*. This entails using the non-semantic low-level cues over $t_{\mathrm{obs}}$ to regress the same cues over the future window $t_{\mathrm{fut}} \coloneqq [f1 \ldots fT]$. In this example we depict the cues of head pose (solid normal), body pose (hollow normal), and speaking status (speaker in orange). The hypothetical uncertainty estimates over $t_{\mathrm{fut}}$ are also depicted as shaded spreads.

In this work, we take an initial step towards a bottom-up approach to forecasting human behavior for free standing conversational groups. Our guiding motivation is to learn predictive representations of general future social behavior by utilizing unlabeled streams of low-level behavioral features. We do this by regressing future sequences of these features from observed sequences of the same features in a self-supervised manner. We term this task of non-semantic future behavior forecasting as *Social Cue Forecasting* (SCF).

Our approach is built on the observation that the *social signal* [17]—the high-level attitudes and social meaning transferred in interactions—is already embedded in the low-level cues [18]. To conceptually illustrate the contrasting top-down and bottom-up approaches on an example task, Figure 1 depicts an instance of a group leaving event in a naturalistic social conversation. Evidence suggests that such events can be anticipated from certain preceding *rituals* [15] reflected in the postural changes of conversing members [1]. van Doorn [15] built a predictor using 200 instances of group leaving found in over 90 minutes of mingling interaction and hand-crafted features. In contrast, our bottom-up approach would entail learning task agnostic representations of future behavior using the entire 90 minutes of data, and then training simpler predictors for group leaving using the learnt representations as input. The figure also illustrates the complexity of naturalistic interactions where cross-group social influence exists. In this work we focus on the simpler setting of a single group in a scene.

There are several challenges intrinsic to computationally modeling future behavior in social conversations. The future is intrinsically uncertain, the forecasts for interaction partners are inter-dependent, and the social dynamics is unique for each grouping of individuals. We address these through the following contributions:

- We formalize the task of SCF. We characertize the modeling challenges involved, and cast the problem into the meta-learning paradigm, allowing for data-efficient generalization to unseen groups at evaluation without learning group-specific models.
- We propose and evaluate two socially aware Sequence-to-Sequence (Seq2Seq) models within the Neural Process (NP) family [19] for SCF in social conversations. Our method encodes complex social dynamics informative of future group behavior into extractable representations for each individual.

This paper is organized as follows. In Section 2 we formally define and characterize the task of SCF. We situate this work within broader literature in Section 3, and review background concepts in Section 4. We propose the Social Process models in Section 5 and describe our experiments in Section 6, concluding with a discussion of our findings in Section 7.

## 2 Social Cue Forecasting

The objective of SCF is to predict future behavioral cues of *all* people involved in a social encounter given an observed sequence of their behavioral features. More formally, let us denote a window

of observed timesteps as $t_{\text{obs}} := [o1, o2, ..., oT]$, and an unobserved future time window as $t_{\text{fut}} := [f1, f2, ..., fT]$, $f1 > oT$. Note that $t_{\text{fut}}$ and $t_{\text{obs}}$ are typically non-overlapping, can be of different lengths, and $t_{\text{fut}}$ need not immediately follow $t_{\text{obs}}$. Given a set of $n$ interacting participants, let us denote their social cues over a $t_{\text{obs}}$ and $t_{\text{fut}}$ respectively as

$$\boldsymbol{X} := [\boldsymbol{b}_t^i; t \in \boldsymbol{t}_{\text{obs}}]_{i=1}^n, \quad \boldsymbol{Y} := [\boldsymbol{b}_t^i; t \in \boldsymbol{t}_{\text{fut}}]_{i=1}^n. \tag{1a, b}$$

The vector $\boldsymbol{b}_t^i$ encapsulates the multimodal cues of interest from participant $i$ at time $t$. These can include head and body pose, speaking status, facial expressions, gestures, and verbal content—any information stream that combine to transfer social meaning.

In its simplest form, given an $\boldsymbol{X}$, the objective of SCF is to learn a single function $f$ such that $\boldsymbol{Y} = f(\boldsymbol{X})$. However, an inherent challenge in forecasting behavior is that an observed sequence of interaction does not have a deterministic future and can result in multiple socially valid ones—a window of overlapping speech between people both may and may not result in a change of speaker [12, 20], a change in head orientation may continue into a sweeping glance across the room or a darting glance stopping at a recipient of interest [21]. In some cases certain observed behaviors—intonation and gaze cues [5, 13] or synchronization in speaker-listener speech [22] for turn-taking—might make some outcomes more likely than others. Given that there are both supporting and challenging arguments for how these observations influence subsequent behaviors [22, p. 5; 13, p. 22], it would be beneficial if a data-driven model expresses a measure of uncertainty in its forecasts. We do this by modeling the distribution over possible futures $p(\boldsymbol{Y}|\boldsymbol{X})$ rather than forecasting a single future.

Another design consideration arises from a defining characteristic of focused interactions—the participants' behaviors are interdependent. Participants in a group sustain equal access to the shared interaction space through cooperative maneuvering [1, p. 220]. Moreover, when multiple groups are co-located, outsiders unengaged in these intra-group maneuvers may also influence the behavior of those within the group [23, p. 91; 1, p. 233], sometimes causing them to leave (see Figure 1). It is therefore essential to capture uncertainty in forecasts at the *global* level—jointly forecasting one future for all participants at a time, rather than at a *local* output level—one future for each individual independent of the remaining participants' futures.

How participants coordinate their behaviors is a function of several individual factors [24, Chap. 1; 1, p. 237]. Consequently, the social dynamics guiding an interaction also has unique attributes for every unique grouping of individuals. Rather than learning group-specific models to capture these unique dynamics, we formulate the forecasting problem in terms of meta-learning, or *few-shot* function estimation. We interpret each unique group of individuals as the meta-learning notion of a task. The core idea is that we can learn to predict a distribution over futures for a target sequence $\boldsymbol{X}$ having captured the group's unique behavioral tendencies from a context set $C$ of their observed-future sequences. We can then generalize to unseen groups at evaluation by conditioning on a short observed slice of their interaction. We believe that this approach is especially suitable for social conversation forecasting—a setting that involves a limited data regime where good uncertainty estimates are desirable. Note that when conditioning on context is removed ($C = \varnothing$), we simply revert to the formulation $p(\boldsymbol{Y}|\boldsymbol{X})$.

## 3 Related Work

Free-standing conversations are an example of what social scientists call *focused interactions*, said to arise when a "group of persons gather close together and openly cooperate to sustain a single focus of attention, typically by taking turns at talking" [23, p. 24]. A long-standing topic of study has been the systematic organization of turn-taking [25–27], with a particular interest in the event of upcoming speaking turns [5–8]. There has also been some interest in the forecasting task itself, to anticipate disengagement from an interaction [9, 15], the splitting or merging of groups [28], the time-evolving size of a group [29] or semantic social action labels [30, 31]. Most of these works use heuristics, either to generate semantic labels [9], model the dynamics itself [29], or hand-craft features [15].

Although not a forecasting task, the closest work that shares our motivation in predicting non-semantic low-level features is the recently introduced task of Social Signal Prediction (SSP) [32]. The objective is to predict the social cues[1] of a target person using cues from the communication partners as

---

[1] In the domain of Social Signal Processing, a *social signal* [17] refers to the relational attitudes displayed by people. It is a high-level construct resulting from the perception of cues (see Fig. 1 in [18]). From this

input (Joo et al. focus on predictions within the same time window [32, Eq. 6]). While the most general formulation of SSP involves forecasting a single timestep for a target person given the entire group's past behavior [32, Eq. 3], generalizing this formulation runs into an inherent problem; applying the definition to forecasting entails iteratively treating each individual as target, learning separate functions for every person. However, as we discuss in Section 2, these futures of interacting individuals are not independent given observed group behavior. Furthermore, a constrained definition of forecasting that predicts an immediate step into the future is limiting, since forecasting an event that occurs after a delay (e.g. a time lagged synchrony [33] or mimicry [14] episode) might be of interest. Operationalizing this definition would entail a sliding window iteratively using predictions over the offset between $t_{\text{obs}}$ and $t_{\text{fut}}$ as input, which would cascade prediction errors.

A related social setting where forecasting has been of interest is that of *unfocused interactions*. These occur when individuals find themselves by circumstance in the immediate presence of each other, such as pedestrians walking in proximity. Early approaches for forecasting pedestrian trajectories were heuristic based, involving hand-crafted energy potentials to describe the influence pedestrians have on each other [34–41]. More recent approaches encode the relative positional information directly into a neural architecture [42–46].

In a broad sense, the self-supervised learning aspects of this work has some overlap with recent approaches focusing on the non-interaction task of visual forecasting. These works have taken a non-semantic approach to predict low level pixel-based features or intermediate representations [38, 47–52], and demonstrated a utility of the learned representation for other tasks like semi-supervised classification [53], or training agents in immersive environments [54].

## 4 Preliminaries

**Meta-learning.** A supervised learning algorithm can be viewed as a function mapping a dataset $C := (\boldsymbol{X}_C, \boldsymbol{Y}_C) := \{(\boldsymbol{x}_i, \boldsymbol{y}_i)\}_{i \in [N_C]}$ to a predictor $f(\boldsymbol{x})$. Here $N_C$ is the number of datapoints in $C$, and $[N_C] := \{1, \ldots, N_C\}$. The key idea of meta-learning is to learn the learning process itself, modeling this function representing the initial algorithm using another supervised learning algorithm; hence the name *meta*-learning. In meta-learning literature, a *task* refers to each dataset in a collection $\mathcal{M} := \{\mathcal{T}_i\}_{i=1}^{N_{\text{tasks}}}$ of related datasets [55]. For each task $\mathcal{T}$, a meta-learner is episodically trained to fit a subset of target points $D := (\boldsymbol{X}, \boldsymbol{Y}) := \{(\boldsymbol{x}_i, \boldsymbol{y}_i)\}_{i \in [N_D]}$ given another subset of context observations $C$. At meta-test time, the resulting predictor $f(\boldsymbol{x}, C)$ uses the information obtained during meta-learning to make predictions for unseen target points conditioned on context sets unseen at meta-training.

**Neural Processes** Sharing the same core motivations, NPs are a family of latent variable models that extend the idea of meta-learning to situations where uncertainty in the predictions $f(\boldsymbol{x}, C)$ are desirable. They do this by meta-learning a map from datasets to stochastic processes, estimating a distribution over the predictions $p(\boldsymbol{Y}|\boldsymbol{X}, C)$. To capture this distribution, NPs model the conditional latent distribution $p(\boldsymbol{z}|C)$ from which a task representation $\boldsymbol{z} \in \mathbb{R}^d$ is sampled. This constitutes the model's *latent* path. The context can also be incorporated through a *deterministic* path, via a representation $\boldsymbol{r}_C \in \mathbb{R}^d$ aggregated over $C$. An observation model $p(\boldsymbol{y}_i|\boldsymbol{x}_i, \boldsymbol{r}_C, \boldsymbol{z})$ then fits the target observations in $D$. The generative process for the NP is written as

$$p(\boldsymbol{Y}|\boldsymbol{X}, C) := \int p(\boldsymbol{Y}|\boldsymbol{X}, C, \boldsymbol{z})p(\boldsymbol{z}|C)d\boldsymbol{z} = \int p(\boldsymbol{Y}|\boldsymbol{X}, \boldsymbol{r}_C, \boldsymbol{z})q(\boldsymbol{z}|\boldsymbol{s}_C)d\boldsymbol{z}, \qquad (2)$$

where $p(\boldsymbol{Y}|\boldsymbol{X}, \boldsymbol{r}_C, \boldsymbol{z}) := \prod_{i \in [N_D]} p(\boldsymbol{y}_i|\boldsymbol{x}_i, \boldsymbol{r}_C, \boldsymbol{z})$. The latent $\boldsymbol{z}$ is modeled by a factorized Gaussian parameterized by $\boldsymbol{s}_C := f_s(C)$, with $f_s$ being a deterministic function invariant to order permutation over $C$. When the conditioning on context is removed ($C = \varnothing$), we have $q(\boldsymbol{z}|\boldsymbol{s}_\varnothing) := p(\boldsymbol{z})$, the zero-information prior on $\boldsymbol{z}$. $C$ is encoded on the deterministic path using a function $f_r$ similar to $f_s$, so that $\boldsymbol{r}_C := f_r(C)$. In practice this is implemented as $\boldsymbol{r}_C = \sum_{i \in [N_C]} \text{MLP}(\boldsymbol{x}_i, \boldsymbol{y}_i)/N_C$. The observation model is referred to as the *decoder*, and $q, f_r, f_s$ comprise the *encoders*. The parameters of the NP are learned for random subsets $C$ and $D$ by maximizing the evidence lower bound (ELBO)

$$\log p(\boldsymbol{Y}|\boldsymbol{X}, C) \geq \mathbb{E}_{q(\boldsymbol{z}|\boldsymbol{s}_D)}[\log p(\boldsymbol{Y}|\boldsymbol{X}, C, \boldsymbol{z})] - \mathbb{KL}(q(\boldsymbol{z}|\boldsymbol{s}_D)||q(\boldsymbol{z}|\boldsymbol{s}_C)). \qquad (3)$$

perspective, the task of Social Signal Prediction [32] is a misnomer since it still relates to the prediction of cues and not signals, a distinction we preserve in this work.

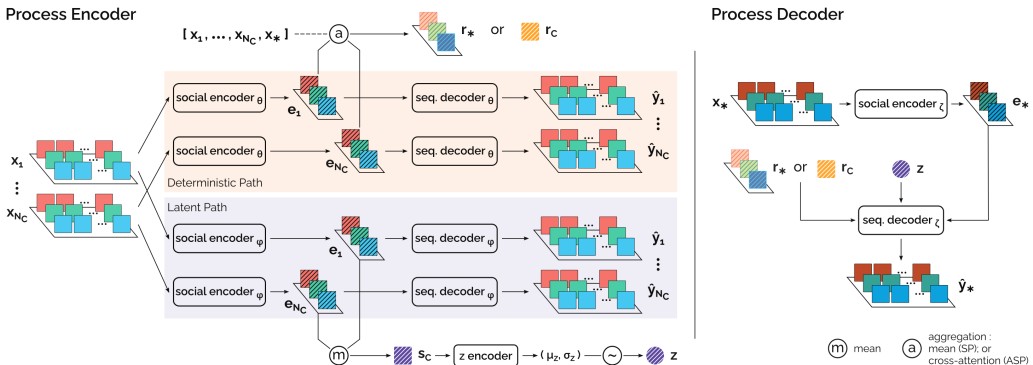

**Figure 2:** Architecture of the SP and ASP family.

## 5 Social Processes

In this section we present our socially aware Seq2Seq models within the NP family that is agnostic to group member identities and group size. To setup the task, we split the contextual interaction on which we condition into pairs of observed and future sequences, writing the context as $C \coloneqq (\boldsymbol{X}_C, \boldsymbol{Y}_C) \coloneqq (\boldsymbol{X}_j, \boldsymbol{Y}_k)_{(j,k) \in [N_C] \times [N_C]}$, where every $\boldsymbol{X}_j$ occurs before the corresponding $\boldsymbol{Y}_k$. As discussed in Section 3, domain experts focusing on behavior analysis might be interested in settings where $\boldsymbol{t}_{\mathrm{obs}}$ and $\boldsymbol{t}_{\mathrm{fut}}$ are offset by an arbitrary delay. Consequently, the $j$th $\boldsymbol{t}_{\mathrm{obs}}$ can have multiple associated $\boldsymbol{t}_{\mathrm{fut}}$ windows. Denoting the set of target window pairs as $D \coloneqq (\boldsymbol{X}, \boldsymbol{Y}) \coloneqq (\boldsymbol{X}_j, \boldsymbol{Y}_k)_{(j,k) \in [N_D] \times [N_D]}$, our focus in the rest of this work is to model the distribution $p(\boldsymbol{Y}|\boldsymbol{X}, C)$.

The generative process for our model we call the Social Process (SP) follows Eq. 2, which we extend to social forecasting in two ways. We embed an observed sequence $\boldsymbol{x}$ for an individual into a condensed encoding $\boldsymbol{e} \in \mathbb{R}^d$ that is then decoded into the future sequence using a Seq2Seq architecture [56, 57]. Our intuition is that this would cause the representation to encode *temporal* information about the future. Further, for every individual we model this $\boldsymbol{e}$ as a function of their own behavior, and that of their partners as viewed by them. The intuition is that this captures the *spatial* influence partners have on the participant over the $\boldsymbol{t}_{\mathrm{obs}}$. Using notation we established in Section 2, we define the observation model for the SP for a single participant $\mathrm{p}_i$ as

$$p(\boldsymbol{y}^i|\boldsymbol{x}^i, C, \boldsymbol{z}) \coloneqq p(\boldsymbol{b}^i_{f1}, \ldots, \boldsymbol{b}^i_{fT}|\boldsymbol{b}^i_{o1}, \ldots, \boldsymbol{b}^i_{oT}, C, \boldsymbol{z}) = p(\boldsymbol{b}^i_{f1}, \ldots, \boldsymbol{b}^i_{fT}|\boldsymbol{e}^i, \boldsymbol{r}_C, \boldsymbol{z}). \quad (4)$$

If decoding is carried out in an auto-regressive manner, we can further write the right hand side of Eq. 4 as $\prod_{t=f1}^{fT} p(\boldsymbol{b}^i_t|\boldsymbol{b}^i_{t-1}, \ldots, \boldsymbol{b}^i_{f1}, \boldsymbol{e}^i, \boldsymbol{r}_C, \boldsymbol{z})$. Following the standard NP setting, we implement the observation model as a set of Gaussian distributions factorized over time and feature dimensions. We also incorporate the cross-attention mechanism from the Attentive Neural Process (ANP) [58] to define the variant Attentive Social Process (ASP). Following Eq. 4 and the definition of the ANP, the corresponding observation model of the ASP for a single participant is defined as

$$p(\boldsymbol{y}^i|\boldsymbol{x}^i, C, \boldsymbol{z}) = p(\boldsymbol{b}^i_{f1}, \ldots, \boldsymbol{b}^i_{fT}|\boldsymbol{e}^i, r^*(C, \boldsymbol{x}^i), \boldsymbol{z}). \quad (5)$$

Here each target query sequence $\boldsymbol{x}^i_*$ attends to the context sequences $\boldsymbol{X}_C$ to produce a query-specific representation $\boldsymbol{r}_* \coloneqq r^*(C, \boldsymbol{x}^i_*) \in \mathbb{R}^d$. The model architectures are illustrated in Figure 2.

**Encoding Partner Behavior.** While a typical Seq2Seq setup conditions the sequence decoder on solely a compact representation of the observed sequence, we'd like to condition an individual's forecast on the observed behavior of both, themselves and their partners. We do this using a pair of sequence encoders: one to encode the temporal dynamics of participant $\mathrm{p}_i$'s features, $\boldsymbol{e}^i_{\mathrm{self}} = f_{\mathrm{self}}(\boldsymbol{x}_i)$, and another to encode the dynamics of a transformed representation of the features of $\mathrm{p}_i$'s partners, $\boldsymbol{e}^i_{\mathrm{partner}} = f_{\mathrm{partner}}(\psi(\boldsymbol{x}_{j,(j \neq i)}))$. Using a separate network to encode partner behavior grants the practical advantage of being able to sample an individual's and partners' features at different sampling rates.

How do we model $\psi(\boldsymbol{x}_j)$? We want the partners' representation to possess two properties: *permutation invariance*—changing the order of the partners should not affect the representation; and *group size independence*—we want to compactly represent all partners independent of the group size.

Beyond coordinate space invariance, we wish to intuitively capture a view of the interaction from $p_i$'s perspective. We extend the approach Qi et al. [59] applied to point clouds to focused interactions by computing pooled embeddings of relative behavioral features. Since most commonly considered nonverbal cues in literature (see Section 6.3) include the attributes of orientation or location (e.g. head/body pose or keypoints) or a binary indicator (such as speaking status), we specify how we transform these. The 3D pose (orientation, location) of every partner $p_j$ is transformed to a frame of reference defined by $p_i$'s pose. At timestep $t$, denoting orientation, location, and binary speaking status for $p_i$ as $\boldsymbol{b}_t^i = [\mathbf{q}^i; \mathbf{l}^i; \mathbf{s}^i]$, and those for $p_j$ as $\boldsymbol{b}_t^j = [\mathbf{q}^j; \mathbf{l}^j; \mathbf{s}^j]$, we have

$$\mathbf{q}^{rel} = \mathbf{q}^i * (\mathbf{q}^j)^{-1}, \quad \mathbf{l}^{rel} = \mathbf{l}^j - \mathbf{l}^i, \quad \mathbf{s}^{rel} = \mathbf{s}^j - \mathbf{s}^i. \tag{6a-c}$$

Note that we use unit quaternions (denoted $\mathbf{q}$) for representing orientation due their various benefits over other representations of rotation [60, Sec. 3.2]. The operator $*$ denotes the Hamilton product of the quaternions. These transformed features for each $p_j$ are encoded using an *embedder* MLP. The outputs are concatenated with $\boldsymbol{e}_{\text{self}}^j$ and processed by a *pre-pooler* MLP, which is followed by the symmetric element-wise Max-pooling function to obtain $\psi(\boldsymbol{x}^j)$ at each timestep. We capture the dynamics in the pooled representation over $\boldsymbol{t}_{\text{obs}}$ using $f_{\text{partner}}$. Finally, we combine $\boldsymbol{e}_{\text{self}}^i$ and $\boldsymbol{e}_{\text{partner}}^i$ for $p_i$ through a linear projection (defined by a weight matrix $W$) to obtain the individual's embedding $\boldsymbol{e}_{\text{ind}}^i = W.[\boldsymbol{e}_{\text{self}}^i; \boldsymbol{e}_{\text{partner}}^i]$. Our intuition is that with information about both $p_i$ themselves, and of $p_i$'s partners from $p_i$'s point-of-view, $\boldsymbol{e}_{\text{ind}}^i$ now contains the information required to predict $p_i$'s future behavior.

**Encoding Future Window Offset.** As we've discussed at the start of this section, a single $\boldsymbol{t}_{\text{obs}}$ might have multiple associated $\boldsymbol{t}_{\text{fut}}$ windows at different offsets. Our intuition is that training a sequence decoder to decode the same $\boldsymbol{e}_{\text{ind}}^i$ into multiple sequences (corresponding to the multiple $\boldsymbol{t}_{\text{fut}}$) in the absence of any timing information might cause an averaging effect in either the decoder or the information encoded in $\boldsymbol{e}_{\text{ind}}^i$. One way around this would be to start decoding one timestep following the end of $\boldsymbol{t}_{\text{obs}}$ and discard the predictions in the gap between $\boldsymbol{t}_{\text{obs}}$ and $\boldsymbol{t}_{\text{fut}}$. However, if decoding is done auto-regressively this might lead to cascading errors over the gap. Instead, we address this one-to-many issue by injecting the offset information into $\boldsymbol{e}_{\text{ind}}^i$ so that the decoder receives a unique encoded representation for every $\boldsymbol{t}_{\text{fut}}$ to decode over. We do this by repurposing the idea of sinusoidal positional encodings [61] to encode offsets rather than relative positions in sequences. For a given $\boldsymbol{t}_{\text{obs}}$ and $\boldsymbol{t}_{\text{fut}}$, and $d_e$-dimensional $\boldsymbol{e}_{\text{ind}}^i$ we define the offset as $\Delta t = f1 - oT$, and the corresponding offset encoding $OE_{\Delta t}$ as

$$OE_{(\Delta t, 2m)} = \sin(\Delta t / 10000^{2m/d_e}), \quad OE_{(\Delta t, 2m+1)} = \cos(\Delta t / 10000^{2m/d_e}). \tag{7a, b}$$

Here $m$ refers to the dimension index in the encoding. We finally compute the representation $\boldsymbol{e}^i$ for Eqs. 4 and 5 as

$$\boldsymbol{e}^i = \boldsymbol{e}_{\text{ind}}^i + OE_{\Delta t}. \tag{8}$$

**Auxiliary Loss Functions.** We incorporate a geometric loss function that improves performance in pose regression tasks. For $p_i$ at time $t$, given the ground truth $\boldsymbol{b}_t^i = [\mathbf{q}; \mathbf{l}; \mathbf{s}]$, and the predicted mean $\hat{\boldsymbol{b}}_t^i = [\hat{\mathbf{q}}; \hat{\mathbf{l}}; \hat{\mathbf{s}}]$, we denote the tuple $(\boldsymbol{b}_t^i, \hat{\boldsymbol{b}}_t^i)$ as $B_t^i$. We then have the location loss in Eucliden space $\mathcal{L}_{\text{l}}(B_t^i) = \left\| \mathbf{l} - \hat{\mathbf{l}} \right\|$, and we can regress the quaternion values using

$$\mathcal{L}_{\text{q}}(B_t^i) = \left\| \mathbf{q} - \frac{\hat{\mathbf{q}}}{\|\hat{\mathbf{q}}\|} \right\|. \tag{9}$$

Kendall and Cipolla [60] show how these losses can be combined using the homoscedastic uncertainties in position and orientation, $\hat{\sigma}_{\text{l}}^2$ and $\hat{\sigma}_{\text{q}}^2$:

$$\mathcal{L}_{\sigma}(B_t^i) = \mathcal{L}_{\text{l}}(B_t^i) \exp(-\hat{s}_{\text{l}}) + \hat{s}_{\text{l}} + \mathcal{L}_{\text{q}}(B_t^i) \exp(-\hat{s}_{\text{q}}) + \hat{s}_{\text{q}}, \tag{10}$$

where $\hat{s} := \log \hat{\sigma}^2$. Using the binary cross-entropy loss for speaking status $\mathcal{L}_{\text{s}}(B_t^i)$, we have the overall auxiliary loss over $t \in \boldsymbol{t}_{\text{fut}}$:

$$\mathcal{L}_{\text{aux}}(\boldsymbol{Y}, \hat{\boldsymbol{Y}}) = \sum_i \sum_t \mathcal{L}_{\sigma}(B_t^i) + \mathcal{L}_{\text{s}}(B_t^i). \tag{11}$$

The parameters of the SP and ASP are trained by maximizing the ELBO in Eq. 3 and minimizing this auxiliary loss function for each of our sequence decoders.

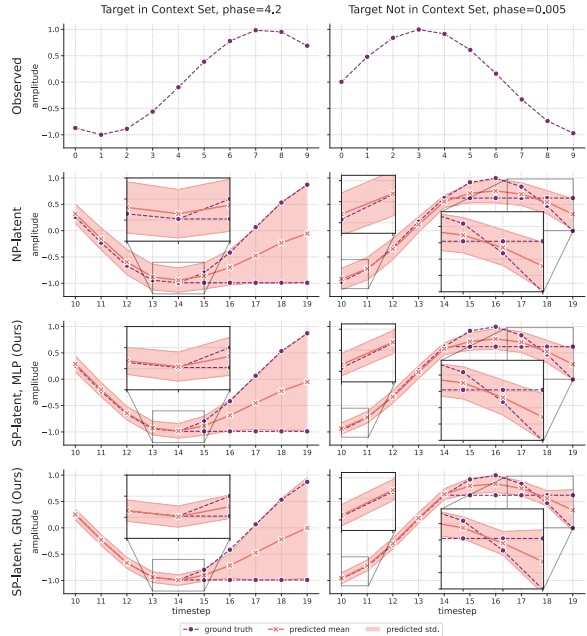

**Figure 3:** Ground truths and model predictions for the toy task simulating the forecasting of glancing behavior.

**Table 1: Mean (Std.) Negative Log-Likelihood (NLL) on the Haggling Test Sets.** The reported mean and std. are over individual sequences in the test sets. Lower is better. The superscript * indicates best NLL within family, boldface best overall.

| | Context | |
| --- | --- | --- |
| | **Random** | **Fixed-Initial** |
| **Baselines** | | |
| NP-latent | 38.34 (19.1) | 37.64 (18.1) |
| NP-latent+det | 40.41 (23.9) | 40.15 (23.0) |
| ANP-dot | 35.66* (20.8) | 38.06* (20.6) |
| ANP-multihead | 40.60 (19.2) | 41.11 (19.2) |
| **Ours (MLP)** | | |
| SP-latent | −74.06 (6.0) | −74.19 (5.9) |
| SP-latent+det | −77.49 (7.8) | −76.90 (8.4) |
| ASP-dot | −76.33 (6.5) | −75.15 (6.5) |
| ASP-multihead | −**83.77*** (**10.3**) | −**83.43*** (**9.7**) |
| **Ours (GRU)** | | |
| SP-latent | −4.23 (27.4) | −3.72 (30.7) |
| SP-latent+det | −17.38* (50.5) | −16.08* (52.2) |
| ASP-dot | 19.91 (46.7) | 31.39 (77.0) |
| ASP-multihead | −7.11 (26.9) | −0.51 (28.8) |

## 6 Experiments and Results

### 6.1 Models and Baselines

Our modeling assumption is that the underlying stochastic process generating the behaviors does not evolve over time. Stated differently, we assume that the individual factors determining how participants coordinate behaviors—age, cultural background, personality variables [24, Chap. 1; 1, p. 237]—are likely to remain the same over the short duration of a single interaction. This is in contrast to a related line of work that deals with *meta-transfer learning*, where the stochastic process itself changes over time [62–65]. We therefore compare against the NP and ANP family which share our model assumptions and meta-learning attributes. Note that in contrast to our methods, these baselines have direct access to the future sequences in the context, and therefore constitute a strong baseline. We consider two variants: *-latent* denoting only the latent path; and *-latent+det*, containing both deterministic and stochastic paths. We further consider two attention mechanisms for the cross-attention module: *-dot* with dot attention, and *-multihead* with wide multi-head attention [58]. We operationalize the original definitions of the baseline models to sequences by collapsing the timestep and feature dimensions. While the ANP-RNN model [66] shares our model assumptions, it is defined for a task analogous to SSP for concurrent car locations, and cannot be operationalized to forecasting in any simple way (see Section 3 discussing the distinction). We experiment with two choices of architectures for the sequence encoders and decoders in our proposed models: multi-layer perceptrons (MLP), and Gated Recurrent Units (GRU). Implementation and training details for our experiments can be found in Appendix C.

### 6.2 Evaluation on Synthesized Behavior: Forecasting Glancing Behavior

With limited behavioral data availability, a common practice in the domain is to train and evaluate methods on synthesized behavior dynamics [31, 67]. In keeping with this practice, we construct a synthesized dataset simulating two glancing behaviors in social settings [21]. We use a 1D sinusoid to represent horizontal head rotation over 20 timesteps. The sweeping *Type I* glance is represented by a pristine sinusoid, while the gaze fixating *Type III* glance is denoted by clipping the amplitude for the last six timesteps. The task is to forecast the signal over the last 10 timesteps ($t_{\text{fut}}$) by observing the first 10 ($t_{\text{obs}}$). Consequently, the first half of $t_{\text{fut}}$ is certain, while uncertainty over the last half results from every observed sinusoid having two ground-truths. It is impossible to infer from an observed sequence alone if the head rotation will stop partway through the future. We describe

additional data setup, model details, and quantitative results for this setting in Appendices A.1, C and D.1, respectively. Figure 3 illustrates the ground truths, predicted means and std. deviations for a sequence within and outside the context set. We observe that all models estimate the mean reasonably well, although our proposed SP models learn a slightly better fit. More crucially, the SP models—especially the SP-GRU—learn much better uncertainty estimates over the certain and uncertain parts of the future compared to the NP baseline.

## 6.3  Real-World Behavior: The Haggling Dataset

We also evaluate our models on real-world behavior data, using the Haggling dataset of triadic interactions [32]. Participants are engaged in an unscripted game where two sellers compete to sell a fictional product to a buyer who has to choose between the two. We use the same split of 79 training sets (groups) and 28 test sets used by Joo et al. [32]. In our experiments we consider the following social cues: *head pose* described by the 3D location of the nose keypoint and a face normal; *body pose* described by the location of the mid-point of the shoulders and a body normal; and binary *speaking status*. Apart from being the most commonly considered cues in computational analyses of such conversations [68–70], pose and turn taking are found to be crucial in the sustaining of conversation [1, 12, 18]. We specify the dataset preprocessing details in Appendix D.2.

## 6.4  Evaluation

**Context Regimes.**  We evaluate all models on two context regimes: *random*, and *fixed-initial*. The *random* regime follows the standard NP setting that the models are trained in. Context samples (sequence-pairs) are selected as a random subset of target samples, so the model is exposed to behaviors from any phase of the interaction lifecycle. Here we ensure that batches contain unique $t_{obs}$ to prevent any single observed sequence from dominating the aggregation of representations over the context split. At evaluation, we take $50\%$ of the batch as context. In the *fixed-initial* context regime, we investigate how the model can generalize knowledge of group specific characteristics from observing the initial dynamics of an interaction where certain gestures and patterns are more distinctive [1, Chap. 6]. This matches what a social agent might face in a real-world scenario. Here we treat the first $20\%$ of the entire interaction as context, treating sequences from the rest as target.

**Evaluation Metrics.**  We report the negative log-likelihood (NLL) $-\log p(\boldsymbol{Y}|\boldsymbol{X}, C)$ in Table 1 (computed by summing over feature dimensions and people, and averaging over timesteps). Beyond the NLL, we also report the error in the predicted means over test sequences in Table 2: mean-squared error (MSE) for the head and body keypoint locations; mean absolute error (MAE) in orientation in degrees; and speaking status accuracy. Note that while the ground truth orientation normals are constrained in the horizontal plane, we don't constrain our predicted quaternions. We therefore report the absolute error in rotation in 3D. The reported mean and std. deviation of all metrics are over sequences in the test sets. We further report the metrics for every timestep over $t_{fut}$ in Appendix A.2, and qualitative visualizations of the forecasts in Appendix B.

## 6.5  Ablations

**Encoding Partner Behavior.**  Modeling the interaction from the perspective of each individual is a central idea in our approach. We investigate the influence of encoding partner behavior into individual representations $r_{ind}^i$ on the performance. We train the SP-latent+det GRU variant in two configurations: *no-pool*, where we do not encode any partner behavior; and *pool-oT* where we pool over partner representations only at the last timestep (similar to [44]). We choose the SP-GRU model since it achieves the best trade-off between minimizing NLL and forecasting cues consistent with human behavior. Both configurations lead to worse NLL and location errors (Appendix A.3).

**Deterministic Decoding and Social Encoder Sharing.**  Error gradients can flow back into our sequence encoders through two paths: from the final stochastic sequence decoder, as well as the deterministic decoders on the latent and deterministic paths. We investigate the effect of the deterministic decoders by training the SP-latent+det GRU model without them. We also investigate sharing a single social encoder between the Process Encoder and Process Decoder in Figure 2. We find that removing the decoders only improves log-likelihood if the encoders are shared, and at the cost of head orientation errors (Appendix A.3).

**Table 2: Mean (Std.) Errors in Predicted Means over Sequences in the Haggling Test Sets.** Lower is better for all metrics except for speaking status accuracy. * indicates best measure within family, boldface best overall.

| | Random Context | | | | | Fixed-Initial Context | | | | |
|---|---|---|---|---|---|---|---|---|---|---|
| | Head Loc. MSE (cm) | Body Loc. MSE (cm) | Head Ori. MAE (°) | Body Ori. MAE (°) | Speaking Accuracy | Head Loc. MSE (cm) | Body Loc. MSE (cm) | Head Ori. MAE (°) | Body Ori. MAE (°) | Speaking Accuracy |
| **Baselines** | | | | | | | | | | |
| NP-latent | 14.21 (6.5) | 15.06 (6.1) | 16.29 (13.8) | 12.82 (13.7) | 0.787 (0.23) | 13.85 (6.1) | 14.71 (5.7) | 16.22 (14.1) | **12.69*** (**13.9**) | **0.774*** (**0.24**) |
| NP-latent+det | 15.01 (7.3) | 15.97 (7.2) | 17.45 (18.3) | 14.65 (20.0) | 0.715 (0.24) | 15.01 (7.5) | 15.95 (7.5) | 17.26 (15.9) | 14.68 (18.7) | 0.701 (0.24) |
| ANP-dot | **11.86*** (**5.4**) | **12.22*** (**5.5**) | **15.44*** (**13.3**) | **12.56*** (**18.0**) | **0.806*** (**0.23**) | **12.83*** (**5.9**) | **13.26*** (**6.0**) | **16.19*** (**13.7**) | 13.56 (17.8) | 0.717 (0.23) |
| ANP-multihead | 16.36 (7.4) | 17.17 (7.2) | 19.41 (20.4) | 16.02 (22.1) | 0.692 (0.21) | 16.68 (7.9) | 17.43 (7.7) | 19.78 (21.2) | 15.57 (20.3) | 0.682 (0.21) |
| **Ours (MLP)** | | | | | | | | | | |
| SP-latent | 25.58 (10.1) | 26.57* (9.0) | 91.07 (23.9) | 97.09 (22.5) | 0.638 (0.08) | 25.27 (10.0) | 26.33* (8.9) | 91.14 (23.8) | 97.09 (22.5) | 0.640 (0.09) |
| SP-latent+det | 31.99 (8.2) | 36.33 (7.3) | 91.08 (23.9) | 91.36 (23.9) | 0.629 (0.18) | 32.93 (9.4) | 37.16 (8.5) | 91.15 (23.9) | 91.36 (23.9) | 0.633 (0.18) |
| ASP-dot | 27.16 (7.7) | 31.19 (7.1) | 90.88 (23.9) | 91.43 (23.8) | 0.704 (0.19) | 27.94 (7.8) | 31.83 (7.1) | 90.93 (23.9) | 91.43 (23.8) | 0.628 (0.20) |
| ASP-multihead | 23.88* (7.8) | 27.13 (7.7) | 90.50* (23.9) | 91.04* (24.1) | 0.792* (0.24) | 24.07* (8.1) | 27.35 (8.3) | 90.53* (23.9) | 91.07* (24.1) | 0.770* (0.25) |
| **Ours (GRU)** | | | | | | | | | | |
| SP-latent | 17.18 (6.5) | 17.41 (6.2) | 17.76* (15.8) | 14.78* (20.7) | 0.713 (0.23) | 16.66 (6.2) | 17.17 (6.0) | 17.67* (16.0) | 14.64* (20.3) | 0.705 (0.23) |
| SP-latent+det | 15.84 (5.5) | 17.76 (7.5) | 20.65 (19.9) | 21.73 (29.5) | 0.671 (0.22) | 16.53* (6.0) | 18.20 (8.0) | 20.74 (19.5) | 21.31 (28.9) | 0.674 (0.22) |
| ASP-dot | 22.49 (8.7) | 22.64 (11.1) | 17.99 (12.8) | 15.58 (19.6) | 0.722 (0.25) | 23.66 (8.7) | 24.50 (11.7) | 19.22 (14.8) | 16.82 (19.4) | 0.620 (0.27) |
| ASP-multihead | 15.18* (6.7) | 15.01* (6.0) | 24.26 (21.3) | 35.06 (38.5) | 0.778* (0.23) | 16.84 (6.9) | 16.80* (6.3) | 25.37 (21.3) | 35.44 (38.0) | 0.725* (0.23) |

# 7 Discussion and Conclusion

What qualifies as the best performing model for SCF? Our SP-GRU learns the best fit for synthesized behavior. On the commonly used metric of NLL [19, 58, 62], our SP-MLP models perform the best for real-world data. However, they fare the worst at estimating the mean. On the other hand, the SP-GRU models estimate a better likelihood than the NP baselines with comparable errors in mean forecast. While the NP baselines attain the lowest errors in predicted means, they also achieve the worst NLL. From the qualitative visualizations and ablations, it seems that the models minimize NLL at the cost of orientation errors; in the case of SP-MLP seemingly by predicting the majority orientation of the two sellers who face the same direction. Also, the NP models forecast largely static futures. In contrast, while being more dynamic, the SP-GRU forecasts also contain some smoothing.

Our synthesized glancing behavior is grounded in social literature, and matches the head pose features in the real-world data (horizontal orientation). Why do we see a large discrepancy in qualitative forecasts? One crucial distinction between the synthetic and real data is the subtlety and sparsity of motion. Our synthesized data makes the common implicit assumption that head pose is a proxy for gaze [31, 67, 68, 70–72]. In real-world data, attention shifts through changes in gaze are not always accompanied by similar head rotations [73, Fig. 5], and gaze is harder to record non-invasively in-the-wild with reasonable accuracy. The consequence of this approximation is exacerbated in the triadic Haggling setting where people are arranged roughly in a triangle and within each other's field of vision, making head movements even more subtle. In natural settings, groups occupy varied formations such as *side-by-side*, or *L-arrangement* [60, p. 213]. Here the more accentuated pose changes could aid in anticipating behavior. From this perspective, the combination of limited data and our simplifying assumption of a single group in a scene is a primary limitation of this work. The only publicly available dataset meeting our assumptions is the Haggling dataset, where all interactions follow similar patterns. As targeted development of techniques for recording such datasets in-the-wild gain momentum [74], evaluating these models in the different interaction settings would yield increased insight. Nevertheless, our aim in evaluating on synthesized as well as real-world data was to highlight the influence that such common implicit assumptions can have on performance when applying methods. As an aside, we believe that this subtlety and sparsity of motion is also an important distinction between forecasting in focused and unfocused interactions. While the same techniques can be applied in both scenarios, pedestrian location is a perpetually changing data stream.

The broader goal of this paper is to take a step towards bridging a gap we perceive between research domains; on one hand, we notice that there is a growing trend of applying deep learning techniques in the small data regime that is social behavior data [30, 75]. Without citing specific works as negative exemplars, this is occasionally accompanied by surface treatment of social science literature. On the other hand, in our conversations we have also perceived a preemptive resistance to deep learning methods precisely due to limited data. We believe that our work here—specifically our conceptualization of conversations groups as meta-learning *tasks* grounded in extensive considerations from social literature; our approach of learning extractable task-agnostic representations of predictive behavior; and the distinction between real-world and synthesized dynamics commonly used for evaluation—is of value in stimulating a broader community discussion about the considerations when applying machine learning approaches within the domain of free-standing social conversations.

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
