# OpenReview forum: "Social Processes: Self-Supervised Forecasting of Nonverbal Cues in Social Conversations"
_NeurIPS.cc/2021/Conference — NeurIPS 2021 Submitted_

### Official Review · Reviewer_JGHq · 2021-07-10

**Rating:** 5
**Confidence:** 3

**Summary:**

The paper addresses the problem of nonverbal social cue forecasting in group social conversations, and proposes a bottom-up self-supervised approach to solve this problem. The bottom-up approach utilizes low-level behaviour cues to predict higher-level  social cues. The authors propose to use meta-learning and neural process to model the social process. Results on simulated and real data show that the proposed method is better than the baselines.

Contributions:
1. The paper takes the first step towards a bottom-up self-supervised method to solve the social cue forecasting problem. Previous methods are all top-down.
2. It utilizes the notion of neural process to formalize the concept of Social Process.


**Ethical Concerns:**

It's worth verifying if the synthetic dataset contains personal-identifiable information like face.

**Ethics Review Area:**

["Privacy and Security (e.g., consent)"]

**Limitations And Societal Impact:**

I think the writing could be improved. The experiments and content look good, but it might be worthwhile working a little bit more on how to organize and present them to the users.  For example, in line 181, "Our intuition is that this would cause the representation to encode temporal information about the future." Is it verified by the experiments?

And also as mentioned above, the results are not described in details.

Also, in Table1 and 4, why would NLL be negative for the models? And why is the difference between models so big? Are statistical tests performed here?

**Main Review:**

Originality: The method is novel since it's the first bottom-up method for predict human behaviour. And the intent of bottom-up is to solve the problem of limited amount of labeled data.

Quality: The paper experimented on both a simulated and a real dataset, and performed necessary ablation studies to show the functionality of each module. But many experimental results are in the appendix.

Clarity: I think the writing could be improved. I found the paper a bit hard to follow because it covers many topics and many different techniques, it feels like the paper is constantly jumping from one topic to another without enough connections between different parts (e.g. why do we need this, and this is our assumption and later we use xx experiment to prove this).  Also, there are no enough descriptions of the experimental results (Table 1 and 2, etc).

Significance: It's not clear if statistical tests were performed from the tables and figures. If the results are statistically significant, I think this work could be a good addition to the current social cue prediction work.







**Needs Ethics Review:**

Yes

**Time Spent Reviewing:**

1 hour

---

> ### Author Response · Authors · 2021-08-10
> **Thank you for the review and questions**
>
> We thank you for the review, and the acknowledgement of the novelty of our method and that we have performed the necessary ablation studies to evaluate individual modules.
>
> 1\. **“‘Our intuition is that this would cause the representation to encode temporal information about the future.’ Is it verified by the experiments?”**
>
> We apologize if this was unclear from the text. Yes, the experiments comparing our models to the NP baselines definitely confirm our intuition.
>
> The intuition is that after end-to-end training, the embedding $\mathbf{e}$ must contain information about the future because it is the only source of information about the observed cues the sequence decoder can access in order to output the future cues.  In contrast, the baseline NP decoders directly access the input sequence to output the future. So our point iis that if the sequence decoders can forecast future cues from only the embedding $\mathbf{e}$ at least as well as a model that directly accesses the input, the embeddings have captured temporal information about the future behavior. From the results in Table 1, since our methods outperform the baselines, we argue that this confirms our intuition.
>
> 2\.  **“Why would NLL be negative for the models?”**
>
> The NLL, just like the log-likelihood, can be both positive and negative for continuous distributions. (Note that it refers to the density rather than the probability of taking on a certain value.) In our case, the log-likelihood is the log of the predicted Gaussian density function evaluated at the ground-truth value of the feature.
>
> 3\. **“And why is the difference between models so big?”**
>
> As we have stated on lines 304-305, we compute the reported NLL ( $- \log p(\mathbf{Y}|\mathbf{X}, C)$ ) by summing over feature dimensions and people in the group, and averaging over the timesteps. So differences can add up so that the overall sum can get large.
> Does the reviewer have a deeper question about where the improved performance comes from? If so, please also see the second half of our response to the next point.
>
> 4\. **“Are statistical tests performed here?”**
>
> We didn’t perform statistical tests; our overarching goal is to show how the bottom-up approach can benefit downstream semantic tasks in the domain through extractable representations and uncertainty estimates. So this ability to have extractable representations from our model is already of value even if the models had performed only as well as the strong NP baselines that directly access the input for their predictions (where having extractable representations is not possible).
>
> However, to provide insight between the differences in model performances, we reported the std. of the metrics over the individual test sequences along with the qualitative visualizations. As for the uncertainty estimates, comparing the SP-GRU and NP models in Tables 1 and 2, and taking the variance over individual sequences into consideration, the predicted means are similar, but the NLLs are different, which implies that our models learn a better std fit over the futures. (This is easier to see in the synthetic 1D case in Figure 3. Also see lines 338-358 for a discussion on differences in qualitative forecasts on synthetic and real-world data.) We can perform the statistical test for added completeness, but our deeper point is that a negative test would not devalue the point of the contribution.
>
> 5\. **“And also as mentioned above, the results are not described in details.”**
>
> Thanks for the suggestion. We will update the captions of the Tables to provide context in a more localized manner next to the results.
>
> 6\. **“It's worth verifying if the synthetic dataset contains personal-identifiable information like face.”**
>
> The synthetic dataset is constructed using 1D sinusoids (lines 268-271, Appendix D.1). The possibility of personal information is therefore not applicable by definition. For the real-world haggling dataset, as specified in Checklist item 4 (e), we have avoided illustrating images of participants to preserve privacy even though the dataset is publicly available. Instead, we have implemented a visualization system in Blender 3D to illustrate the behavioral features in Fig. 9 and 10. The faces in Fig. 1 are blurred for the same reason.

---

> > ### Comment · Reviewer_JGHq · 2021-08-24
> > **Thanks for your response**
> >
> > The authors addressed most of my questions, but I am still concerned about the statistical tests and think it's important to quantitatively show that the improvements didn't come from randomness, so I will keep my score the same. I do encourage the authors to include the test in the next version.

---

> > > ### Author Response · Authors · 2021-09-01
> > > **Statistical tests using reported means and std of the metrics indicate significant performance difference**
> > >
> > > We agree with the reviewer that such numbers would complement our analysis and now carried out an unpaired t-test based on the means and standard deviations that we have already reported in the paper. The main outcome is that the difference between best performing NP model (ANP-dot) and SP model (SP-GRU latent+det) are significantly different, as indicated by a p-value less than 0.0001. (the results are significant even with correcting for multiple-testing). We will gladly add this to the final version of the paper.

---

### Official Review · Reviewer_u6ot · 2021-07-16

**Rating:** 4
**Confidence:** 4

**Summary:**

This work proposes the Social Process (SP) models to predict human behaviors in social conversations.

**Limitations And Societal Impact:**

Some limitations are discussed in the last section of this work.

**Main Review:**

- First of all, I don't have sufficient knowledge to evaluate this work from the perspective of psychology, so most of the following evaluation will focus on the machine learning aspects
- The notations and technical details used in this work are very hard to follow. For example, in the paragraph of "Neural Processes", I assume this paragraph is mainly about variational inference, at least judging from equation 3. However, I quickly got lost in the technical terms, such as
  - Line 162 - 153: "The latent z is modeled by a factorized Gaussian parameterized by $s_C:=f_s(C)$, with $f_s$ being a deterministic function invariant to order permutation over $C$", I have no idea what this line means
  - Line 165: "C is encoded on the deterministic path ...", what is the definition of a deterministic path?
  - Equation 3, what are $q(z|s_D)$ and $q(z|s_C)$ respectively? what are the parameters of this optimization problem?
- In section 5
  - Line 173: "every $X_j$ occurs before the corresponding $Y_k$", are $X_j$ and $Y_k$ pairwised?
  - Line 180: "Our intuition is that this would cause the representation to encode temporal information about the future", I am not sure I agree with this statement. A seq2seq model can learn a mapping between inputs and outputs, but it may not be powerful enough to encode the future
  - The use of positional encoding in the paragraph of "Encoding Future Window Offset" is really out of nowhere. In BERT, the use of positional embedding is to avoid the self-attention mechanism that confuses the word order information. While in sequential modeling, I am not sure it is necessary. At least, from the paper, I did not understand the motivation.


The abovementioned items are just some example issues of this work. Overall, I think this work tries to combine some psychology concepts with machine learning, by throwing out lots of machine learning techniques without sufficient understanding.

**Time Spent Reviewing:**

3

---

> ### Author Response · Authors · 2021-08-10
> **Thank you for the review and questions**
>
> We thank you for the review and the comments on the machine learning aspects of the work. We acknowledge that the interdisciplinary nature of the paper may indeed make it hard to read for those not familiar with the social side. The references provided in the related work section give ample background material to close such gaps. Given the complexity of the topic, bringing together a specialized technical field and a specialized application domain, it is of course impossible for the paper to be completely self-contained.
>
> 1\. **Notation concerns from background literature on Neural Processes.**
>
> **“I assume this paragraph is mainly about variational inference, at least judging from equation 3.”**
>
> More specifically, this part of the Background Section is about the Neural Process family of models that we build upon. We appreciate the comments and the effort spent by the reviewer and, perhaps, we can dedicate more space to explain the notations and jargon in more detail. We would like to point out, however, that the notations and terms we use are largely consistent with what is used in the prior literature (references [19] and [58] in the paper). This is to the benefit of the reader already familiar with the background literature. We did however make the following minor changes over the notation in [58] to maintain internal consistency, and improve notation precision:
> - The subset of target points in our paper is denoted by $D$ instead of $T$ as in [58] to avoid confusion with meta-learning tasks $\mathcal{T}$.
> - The samples in the context and target sets are defined as tuples in [58], e.g. $(\mathbf{x}_C, \mathbf{y}_C)  := (\mathbf{x}_i, \mathbf{y}_i)_\{i \in C\}$ for the context set. We replace this with $C := (\mathbf{X}_C, \mathbf{Y}_C) := \\{(\mathbf{x}_i, \mathbf{y}_i)\\}_\{i \in [N_C]\}$ where $[N_C] := \{1, \ldots, N_C\}$ (line 146) to more explicitly indicate the set of tuples and avoid self referential definition for the indices.
> - The functions $\mathbf{r}$ and $\mathbf{s}$ denoting the encoding networks in [58] are replaced by $f_r$ and $f_s$ to better distinguish between their output vectors $\mathbf{r}_C$ and $\mathbf{s}_C$. We further explain in line 166 how this is commonly implemented in practice as $\mathbf{r}_C = \sum_\{i \in [N_C]\}\mathrm{MLP}(\mathbf{x}_i, \mathbf{y}_i) / N_C$.
>
> While we address the specific questions of the reviewer here for convenience, we note that some of these questions might be resolved by additionally referring to the Neural Process literature for familiarization.
>
> - **“Line 162 - 153: ‘The latent $\mathbf{z}$ is modeled by a factorized Gaussian parameterized by $\mathbf{s}_C$, with $f_s(C)$ being a deterministic function invariant to order permutation over ‘, I have no idea what this line means”**
>
>   By factorized, we mean that the $d$-dimensional latent representation $\mathbf{z}$ is composed of $d$ independent Gaussians (or equivalently, modeled using a diagonal covariance matrix). When computing the representations $\mathbf{s}_C$ (and $\mathbf{r}_C$), the corresponding functions $f_r$ and $f_s$ should be invariant to the order in which the samples from the context set are presented to them. Does this help clarify the matter?
>
>
> - **“Line 165: ‘$C$ is encoded on the deterministic path …’, what is the definition of a deterministic path?”**
>
>   The term is introduced in [58] to refer to the path in the model involving a direct conditioning on the context set to obtain the representation $\mathbf{r}_C$ (also referred to as deterministic connections in [19]). This is in contrast to the latent path where an additional sampling operation is performed to obtain $\mathbf{z}$, introducing stochasticity.
>
> - **"Equation 3, what are $q(\mathbf{z}|\mathbf{s}_D)$ and $q(\mathbf{z}|\mathbf{s}_C)$ respectively? what are the parameters of this optimization problem?”**
>
>   Both $q(\mathbf{z}|\mathbf{s}_D)$ and $q(\mathbf{z}|\mathbf{s}_C)$ are approximate distributions computed over the target set and context set respectively. Eq. 3 in our paper corresponds to Eq. 3 in [58] and Eq. 9 in [19]. We denote the neural network components of the model in line 167. The parameters of these encoders and decoder are the parameters being learnt by maximizing the ELBO.
>
> 2\. **Questions concerning our proposed method (Section 5).**
>
> - **“Line 173: ‘every $\mathbf{X}_j$ occurs before the corresponding $\mathbf{Y}_k$’, are $\mathbf{X}_j$ and $\mathbf{Y}_k$ pairwised?”**
>
>   Yes, $\mathbf{X}_j$ and $\mathbf{Y}_k$ denote a single pair of sequences in our sequence-to-sequence prediction problem. In lines 171-172, we describe that we split the context interaction into pairs of observed and future sequences, rewriting the Context set in our problem setting as $C := (\mathbf{X}_C, \mathbf{Y}_C) := (\mathbf{X}_j, \mathbf{Y}_k)_\{(j, k) \in [N_C] \times [N_C]\}$.
>
>   For example, consider we have a 10 minute context interaction at 30 fps, with an observed sequence length of 60 frames, future length of 60 frames,  a maximum offset between them of 150 frames, and a time stride of 5 frames. We would do this by running a sliding window over the 10 minute interaction, and pairing every j-th observed sequence with all subsequent sequences starting after the end of that sequence ($f1 > oT$ as mentioned on line 73) within the next 150 frames to generate our sequences, so for every $\mathbf{X}_j$ we have more than one corresponding future sequence $\mathbf{Y}_k$.
>
>
> - **“Line 180: ‘Our intuition is that this would cause the representation to encode temporal information about the future’, I am not sure I agree with this statement. A seq2seq model can learn a mapping between inputs and outputs, but it may not be powerful enough to encode the future”**
>
>   We are expressing an intuition that guides our model design, but maybe we can reword it to be clearer. In a sense, if the trained Seq2Seq model can predict the future (outputs) from the observed sequence (inputs), then the encoding $\mathbf{e}$ must contain information about the future: it’s the only source of information that the sequence decoder has about the observed cues in order to forecast the future cues.
>   Further, we argue that later comparing our models against the Neural Process (NP) baselines in our experiments verifies this intuition. The baseline NP decoders directly access the input cues to output the future. So our intuition is that if our sequence decoders can forecast future cues from only the embedding e at least as well as a model that directly accesses the input, the embeddings have captured temporal information about the future behavior. From the results in Table 1, since our methods outperform the baselines, we argue that this confirms our intuition.
>
>
> - **“The use of positional encoding in the paragraph of ‘Encoding Future Window Offset’ is really out of nowhere. In BERT, the use of positional embedding is to avoid the self-attention mechanism that confuses the word order information. While in sequential modeling, I am not sure it is necessary. At least, from the paper, I did not understand the motivation.”**
>
>   Referring to our generation of sequence pairs above (and lines 223-224), we have shown how a single observed sequence $\mathbf{x}^i$ can correspond to multiple future sequences. The sequence encoder encodes the $\mathbf{x}^i$ into the corresponding $\mathbf{e}^i$, which the sequence decoder then decodes. This means the sequence decoder needs some way to know at what point in the future its decoding is to begin. (Note that as we motivate on line 229, if the decoding is autoregressive and done immediately after the end of $\mathbf{x}^i$, the errors may cascade before the decoding reaches the start of the $\mathbf{y}^k$ of interest).
> We would argue that our proposal here is in fact a creative application of the underlying notion of positional encoding to solve a relevant modeling problem. We understand how the “Attention is All You Need” paper uses them to encode the relative location of a word in a sentence. In our problem, we use it to instead encode the offset between the observed and future sequence. We believe that our creative use of an existing solution is a strength rather than a weakness, and allows us to reduce the number of parameters of our model since we do not learn an encoding for the offset.

---

> > ### Comment · Reviewer_u6ot · 2021-08-18
> > **Some concerns addressed**
> >
> > The responses address some concerns from my previous comments and I would like to update my evaluation score up by one point.
> >
> > In addition, the responses also attributed some of the concerns to "the complexity of the topic" and claimed that "it is of course impossible for the paper to be completely self-contained", which I don't think I agree with. Essentially, the major concerns from a modeling perspective are the novelty of the proposed method and the justification of choosing a particular model design.

---

> > > ### Author Response · Authors · 2021-08-18
> > > **Thank you for the comments, additional response**
> > >
> > > Thank you for going through our response, and the comments.
> > >
> > > 1\.   **"..attributed some of the concerns to "the complexity of the topic" and claimed that "it is of course impossible for the paper to be completely self-contained", which I don't think I agree with.**
> > >
> > > By 'self-contained', we were referring to only the precise technical and implementation details from the Neural Process background literature which are too much to fit into the paper beyond what we have included in the section 4 about preliminaries. We agree that the novelty of our proposition and the motivations for the modeling choices should be clearly included. We did try to make the social science motivation clear but can certainly revisit and clarify it further. Any pointers on which parts that were unclear would be greatly appreciated so we can target the problematic parts.
> > >
> > > 2\. **"the major concerns from a modeling perspective are the novelty of the proposed method and the justification of choosing a particular model design."**
> > >
> > > We believe the novelty lies in more than simply the architecture or proposing a seq2seq sub-family within the Neural Process family. Rather, it also lies in bringing a bottom-up representation learning approach in a small-data regime, specifically through the considerations in lines 79-109 grounded in social science literature; these include the choice of modeling uncertainty at the group level rather than individual level, and the motivations for viewing of groups as meta-learning tasks, both of which are novel approaches.
> > >
> > > Given the applied aspects of our paper (which is in keeping with the "Applications (e.g., speech processing, computer vision)" part, and building towards the "Social Aspects of Machine Learning (interpretability)" part of the NeurIPS cfp), we believe that considering the novelty from just the architecture perspective misses a crucial part of the whole picture. Proposing the task itself to exploit unlabeled data and characterizing the approach in Section 2 grounded in the social science literature are also important novel contributions.
> > >
> > > It was not fully clear to us which parts remain unclear. Could you elaborate on which parts (including the choice of encoding of the offset in observed and future sequences) that need further clarity in the justifications? We would be very happy to be able to clarify further.

---

### Official Review · Reviewer_6r9M · 2021-07-16

**Rating:** 5
**Confidence:** 4

**Summary:**

This paper focuses on forecasting the certainty of nonverbal behavior cues for each individual in a group social conversation. Instead of using limited labelled data, this paper uses low-level unlabeled data to build models that estimate the certainty of individuals' future behivour, given their past behivour. This paper proposes Social Process (SP) models, socially aware sequence-to-sequence (Seq2Seq) models within the Neural Process (NP) family, to handle this task. Evaluation on synthesized and real-world behavior data shows that the proposed SP models achieve higher log-likelihood than strong baselines.

**Main Review:**

Originality:
The task proposed in this paper, forecasting he certainty of nonverbal behavior cues in social conversation, is a novel computational task in the context of social science. However, this task similar to language modeling task, a typical NLP task with long-standing research effort, which is to use the history sequence to predict the future sequence. The main different here is that language modeling is to calculate the probaility for the future sequences, while this task is to calculate certainty for the future sequences. In order to calculate certainty, this paper introduces seq2seq-based Nerual Processing model. Starting from this point, most of the contents here are more or less direct applications of existing technologies. For example, the cross-attention mechanism from the Attentive Neural Process (ANP) is used as the backbone of model for single participant; The approach in Qi et al. ([59] in the submission) to process point clouds is used to model partners' representation in the conversation. In summary, the technical contribution for this paper is incremental for NeurIPS. I think this submission may be more appropriate for CVPR or ICCV conference as they are interested in general CV application papers.

Quality & Clarity:
This paper presents a fairly completed work with various experiments, including synthesized and real-world behavior data. The paper is well written and easy to follow. The task requires various of information, including people pose and positions and reproduce this research could be complicated. I suggest the author to release all preprocessed information/features to help other researchers.

Significance:
This paper proposes a task with strong social science motivation. It is excited to see that nerual network systems have been applied to various fields to help domain experts. However, the model proposed in this paper can be viewed as a combination of various machine learning and compter vision technologies. Researchers work for machine learning and neural models may not be interested in this submission. I suggest authors to submit this paper to more general CV conferences (e.g., CVPR, ICCV)

**Time Spent Reviewing:**

4

---

> ### Author Response · Authors · 2021-08-10
> **Thank you for the review**
>
> We thank you for the review, and for acknowledging that we have presented a complete work. We are glad that you found the paper well written and easy to follow.
>
> 1\. **“Researchers work for machine learning and neural models may not be interested in this submission.”**, **“... may be more appropriate for CVPR or ICCV conference as they are interested in general CV application papers.”**
>
> We contend this statement by first quoting the relevant areas and corresponding associated topics from the NeurIPS 2021 Call for Papers that we believe this work directly fits into:
> * Deep Learning (architectures, generative models),
> * Applications (e.g., speech processing, computer vision),
> * Probabilistic Methods (variational inference, Gaussian processes).
>
> Less directly, it is also relevant to
> * Social Aspects of Machine Learning (interpretability) : as we discuss in point 1. in our response to reviewer 29nJ, this work builds towards enabling bottom-up approaches for the asking of interpretability questions pertaining to social human behavior.
>
> Further, as we argue in lines 76-78, our method is agnostic to the modality of the social cues, and is therefore not a core computer vision contribution. The core contributions of  setting up a self-supervised task and a meta-learning approach to get representations and uncertainty estimates in a small-data regime domain are closer to a machine learning contribution.
>
> 2\. **“In summary, the technical contribution for this paper is incremental for NeurIPS.”**
>
> We would like to note that the technical contribution of this paper goes beyond the model proposition. Concretely, the contributions are 1. the setting up and characterization of a self-supervised task, and 2. a meta-learning approach to solve the task in a data efficient manner. These contributions are still technical in our opinion, and pave the way for applying representation learning techniques in the domain, which is not an incremental contribution (See lines 358-368 for a meta discussion). So even if the model we proposed would have achieved a similar performance to baselines, the fact that we allow for extractable representations that can be used for downstream tasks to study future behavior is novel and an important result for this area. As we discussed above in point 1., the novelty of the Seq2Seq Neural Processes is still relevant to the NeurIPS topic area pertaining to architectural improvements, but is only one part of the technical contribution.
>
> 3\. **“However, this task similar to language modeling task, a typical NLP task with long-standing research effort, which is to use the history sequence to predict the future sequence. The main different here is that language modeling is to calculate the probaility for the future sequences, while this task is to calculate certainty for the future sequences.”**
>
> While both language modeling and our proposed task of social cue forecasting involve sequential data, and at first blush they seem similar, there are crucial differences between the domains that make our contributions important:
> * *Availability of data* : As we motivate in the introduction, human behavior datasets are quite small, far smaller than NLP datasets. They are often also collected to study specific behaviors of interest (turn taking, mimicry, etc.) so labeled sequences are even fewer, which is an important modeling challenge.
> * *Nature of data* : As we discuss in Section 7 lines 338-350, changes in the behavioral features over time can be subtle. While discrete tokens in NLP tasks are often passed through an embedding layer to get different representations for each token in a vocabulary, there is no such vocabulary for continuous data, and differences in feature streams across timesteps can be small.
>
> Crucially, our overarching approach has been the opposite of simply taking a deep model from the NLP domain and indiscriminately applying it to some data. Our core modeling choices are motivated through principled grounding in domain considerations: setting up the self-supervised task for data efficiency, setting up groups as meta-learning tasks to efficiently generalise to unseen groups at test. These choices are agnostic of backbone architectures. Moreover, the contribution can go in the reverse direction where our work here is relevant to NLP researchers working in small data regimes such as low-resource languages.
>
> 4\. **“In order to calculate certainty, this paper introduces seq2seq-based Nerual Processing model. Starting from this point, most of the contents here are more or less direct applications of existing technologies. For example, the cross-attention mechanism from the Attentive Neural Process (ANP) is used as the backbone of model for single participant;”**
>
> We would like to point out that our framework is agnostic to architectural choices for the sequence encoders and decoders. As for the mentioned Artificial Neural Process (ANP) variants, our results in Table 1 show that even without the cross-attention module, our Social Process variants achieve far better performance than the baseline models that have direct access to the future sequences in the context set.
>
> 5\. **“I suggest the author to release all preprocessed information/features to help other researchers.”**
>
> We have indeed already provided the features, train/test splits and test batches for reproduction in the supplementary material. Please refer to the README in the code for details. Specifically, the computed features are available at “artefacts / datasets / panoptic-haggling / haggling-hbps.h5”. Should the paper be accepted, we would of course release all the materials provided here.

---

> > ### Comment · Reviewer_6r9M · 2021-09-02
> > **Thank you for your response**
> >
> > The authors addressed part of my concerns, but I still think that the technical contribution for this paper is incremental for NeurIPS, so I will keep my score the same.

---

> > > ### Author Response · Authors · 2021-09-02
> > > **Thank you, request for actionable suggestion**
> > >
> > > We respect the reviewer's decision, but are nonetheless sorry to see that our honest attempts at a response had so little positive effect.  Could we ask the reviewer for one last piece of feedback, and request some actionable suggestions on what we would need to add to push the contribution beyond incremental?

---

### Official Review · Reviewer_29nJ · 2021-07-19

**Rating:** 6
**Confidence:** 3

**Summary:**

This paper presents social process models that solve a social cue forecasting problem. It is defined as predicting figure behavioral cues of participants who have social interaction given observed their behaviors. The social process (SP) is based on a neural process (NP) that can solve the meta-learning problem, and mix the seq2seq models to encode the social behaviors. The authors also suggest encoding a partner’s behaviors. Experiments on the synthetic dataset that mimics glancing behavior show that SP outperforms NP in terms of fitting in the future timestamp. Experiments on a real dataset that captures the head, body, and speaking behaviors of three people during a conversation show that SP outperforms NP in terms of predicted negative log-likelihood.

The main strength of this paper is suggesting an interesting task - social cue forecasting and quite simple models (# parameters 3.0M) to resolve the problem than baseline (NP, # parameters 2.8M).  I enjoy reading this paper since it can make many interesting questions that promise researchers to do future research.


**Limitations And Societal Impact:**

I have some questions and comments that I would like to hear from the authors.

- Can we figure out the important social cues that affect other participants to change their social behaviors in the future time window?

- One of the pros of SP models is being more dynamic in the future sequence, but NP models do not. Then, it would be better to test the predicted means over a different length of the test sequences. I think that SP models outperform NP models when the test sequence is quite long.

- It is hard to understand the examples of qualitative visualizations in appendix B because of the shaking of the bodies (or std area). And I cannot see the difference of social cues from 60 to 111 in the ground truth. It would be better to draw the examples simply or write the text that describes the important social cues in each image.


**Main Review:**

Originality: The task is quite new and the model is also new. SP uses NP and seq2seq, so they are quite well-known models.

Quality: Experiment results show that the suggested model quite performs well to predict behaviors in terms of negative log-likelihood, but baselines (NP and ANP) outperform to predict head orientation and speaking status. They are important to understand communication behaviors since head orientation is related to listening behavior. It is my big concern with this paper since we want to understand human behaviors such as who is talking to whom, not just simulating future behaviors only.

Clarity: This paper is well-written and well organized.

Significance: I am still curious about the results in Table 2. But other researchers are likely to use the author’s ideas and try to solve the social cue forecasting problem.


**Time Spent Reviewing:**

12

---

> ### Author Response · Authors · 2021-08-10
> **Thank you for the review and questions**
>
> We thank you for the review, and are grateful for the time you spent with our manuscript. We are also glad for the acknowledgement that our task formulation and proposed method are novel.
>
> 1\. **“It is my big concern with this paper since we want to understand human behaviors such as who is talking to whom, not just simulating future behaviors only”**, **“Can we figure out the important social cues that affect other participants to change their social behaviors in the future time window?”**
>
> *Summary*: While enabling the asking of these downstream research questions to gain behavioral insight in a bottom-up manner is indeed the broader goal of this work, we believe the stated concern is not a valid limitation of the proposed work. The present proposal pertains to learning extractable representations and uncertainty estimates of non-semantic future behavior in a data-efficient manner, not how they can be used for specific downstream semantic tasks, which are better suited for dedicated future works.
>
> *Longer discussion*: The broader impact of our work here is indeed to enable asking these, and other, downstream semantic questions about future social signals (in lines 43-44 and the footnote on page 3-4, we distinguish between low level cues and high-level signals that are more in-line with what the reviewer is describing). More precisely, rather than understanding the now, the vision is to enable understanding of how the now affects the future behavior.
> The present focus, however, is not to close the loop in addressing any particular open top-down research question about specific behaviors such as “who are the future conversation partners” or “why did a participant change the addressee?”. We believe doing so is better suited to their own dedicated future works that require nuanced consideration (e.g. do passive listeners count as conversation partners?).
> The focus is instead on proposing and evaluating a machine learning methodology that can pave the way for the effective application of representation learning approaches in this small-data regime. We therefore contend that the stated concern is not a valid limitation of the proposed contribution, which is the setting up of a self-supervised task over low-level features (cues), and a meta-learning approach that allows for extractable representations and uncertainty estimates. Not specific techniques to use these encoded representations and estimates to infer intent or explanations, or gain insight into specific behaviors (more related to social signals).
>
>
> 2\. **“Then, it would be better to test the predicted means over a different length of the test sequences. I think that SP models outperform NP models when the test sequence is quite long.”**
>
> We have provided per-timestep mean metrics (Appendix A.2) for our experiments to provide insight in this regard. From Figure 6 we see that even over shorter forecast sequences the GRU backbone SP models still tend to perform better than the baselines on NLL, and  errors in means are comparable with the best performing baseline for all cues for some SP-GRU variants. Combined, this indicates that the SP-GRU learns better uncertainty estimates, which is also verified in the qualitative visualizations (Figures 9 and 10).
>
>
> 3\. **“It would be better to draw the examples simply or write the text that describes the important social cues in each image.”**
>
> Thanks for the suggestion. We assume that you are referring to Figure 9. We shall update that caption to clarify this (there is a subtle leaning motion by the speaker). Would such an addition be sufficient?

---

> > ### Comment · Reviewer_29nJ · 2021-08-22
> > **Thank you for the responses**
> >
> > Thank you for your responses to my questions.
> >
> > 1. I understand that this paper focuses on building future behaviors over time (bottom-up approach as the authors said), not predicting a future behavior only (top-down approach). I believe that the suggested approach is useful for the downstream tasks in terms of increasing interpretability. But it would be better to show the differences between them with experiments results, because the authors insist that they take the first step in the direction of the bottom-up approach.
> >
> > 2. Thanks for the explanation.
> >
> > 3. Yes, Figure 9 is one of the examples. I understand that the results of NP are static but SP generates more dynamic movements. But, I cannot figure out the facing direction even in the observed and ground-truth dataset. Is there any difference among the ground-truth in terms of facing direction? So I suggest that describing the significant events in the timeline. As for me, the results of SP are doing dancing because of the std.

---

> > > ### Author Response · Authors · 2021-09-01
> > > **Thank you for the comments**
> > >
> > > 1\. It may be that we are misunderstanding the reviewer’s request; let us therefore apologize beforehand in case this is indeed so. We really want to understand the suggestion, if only to help us with any subsequent submission. As stated, our work takes “a first step in the direction of a bottom-up approach.” To us it seems that what is asked for is an experiment that somehow shows how a top-down approach compares to ours, being bottom-up. We, however, think this is quite a big ask.  As far as we can see, the experiments and the amount of work to carry out would be rather nontrivial; more at the level of a full-fledged follow-up paper. Maybe the reviewer can give us a precise and workable suggestion of what would be expected?
> > >
> > > ---
> > >
> > > Supporting example of such an experiment:
> > >
> > > One potential source of misunderstanding is that the crucial difference between the top-down and bottom-up approaches is not only that the future behaviors are predicted over a window, but also the type of behaviors. In contrast to the objective low-level cues such as pose and speaking status that we use in our methodology, top-down behaviors are often subjective and typically require manual annotation using a coding scheme designed by domain experts.
> > > For example, one of the design motivations of the Haggling dataset we use in our experiments is “to maximize the influence of each seller’s behavior on the buyer’s decision-making” [1, Sec. 4.1]. One example of a potential downstream forecasting task here would be anticipating the influential behaviors of each seller. A typical top-down approach a domain expert might take is as follows:
> > > - Define an episode of perceived influential behavior grounded in social theory, e.g. using dominance as a proxy for influence, and using the annotation scheme for dominance [2].
> > > - Label windows of change of dominant behavior of each seller.
> > > - Perform a statistical analysis to evaluate the correlation, or causation, between the windows of high dominance of the sellers and preceding group behavior.
> > > Comparing the broad bottom-up approach we propose would involve taking the representations of our models for sequences preceding episodes annotated as dominant behavior, and training a simple method using the representations as input instead of the underlying cues.
> > >
> > > In our current work,  we present the non-trivial task formulation grounded in social theory, and the modelling methodology to formulate such a bottom-up approach.
> > >
> > > [1] Joo, H., Simon, T., Cikara, M., & Sheikh, Y. “Towards social artificial intelligence: Nonverbal social signal prediction in a triadic interaction.”
> > >
> > > [2] Jayagopi, Dinesh Babu, Hayley Hung, Chuohao Yeo, and Daniel Gatica-Perez. "Modeling dominance in group conversations using nonverbal activity cues."
> > >
> > > 3\. The ground truth future involves a subtle rotation of the head by the speaker toward the listener on the left of the image and then back. The speaker also moves closer and lowers their head, bringing it closer to the body (compare frame 111 to frame 60) before the listener on the left starts speaking. We shall update the caption to describe this.

---

### Review · Ethics_Reviewer_VZYf · 2021-08-11

**Recommendation:**

It is possible to address these concerns in the current version of the paper. The paper would benefit from a brief discussion of the possible applications and societal impacts of this research – especially whether it could be used in surveillance.

**Ethical Issues:**

Yes

**Ethics Review:**

Potential negative societal impacts
Better forecasting of nonverbal behavioural/social cues could presumably allow for better surveillance. This is a very ‘upstream’ methodological contribution, rather than more applied future work in which these concerns would arise more intuitively. Nevertheless, the paper would have benefited from some discussion of the possible applications of this methodology, and the societal implications of those applications. This could be positive, such as a social robot (line 18) but could also be used to better predict who is speaking, or even real-time prediction of turn-taking and group-leaving. This therefore touches on ethics subcategory 5 (surveillance) and 4 (if used to improve worker surveillance.

General ethical conduct
As the authors note, there could be some concern about personally-identifiable information in relation to the real-world haggling dataset.

---

> ### Author Response · Authors · 2021-08-17
> **Thank you for the suggestion!**
>
> Thank you for your review, and the acknowledgement that we have considered and addressed the possibility of personally-identifiable information.
>
> 1\. **“This is a very ‘upstream’ methodological contribution, rather than more applied future work in which these concerns would arise more intuitively. Nevertheless, the paper would have benefited from some discussion of the possible applications of this methodology, and the societal implications of those applications.”**
>
> This is a great suggestion. We will expand our discussion of the broader impact of this work on the practice of inter-disciplinary research (in lines 355-368) to include specific examples of down-stream tasks and applications and the associated societal implications.

---

### Review · Ethics_Reviewer_GdrQ · 2021-08-12

**Recommendation:**

This work largely ignored the possible negative societal implications of their work, despite the CFP explicitly asking authors to reflect on these for their work.

**Ethical Issues:**

Yes

**Ethics Review:**

This paper discusses following and classifying human behavior and doesn't discuss the potential harms of such technology, nor how they might be unequally distributed across different demographics.

---

> ### Author Response · Authors · 2021-08-17
> **Thank you for the review and comments**
>
> Thank you for the review and the comments.
>
> 1\. **“This paper discusses following and classifying human behavior, … nor how they might be unequally distributed across different demographics.”**
>
> Note that we aren’t classifying, nor making any claims about, how humans behave; contributions of this nature might be more suitable for downstream research in domains such as psychology or social signal processing. Specifically, as we mention in the introduction (lines 43-44) and in the footnote on page 3-4, in this work we focus on forecasting low-level cues (low-level behavioral features) rather than higher order signals (related to intent or explanation of future behavior). So the consideration of our contribution varying across demographics doesn’t directly apply.
> However, to mitigate future misuse, we will expand the discussion in lines 355-368 to include a reminder to downstream researchers to be cognizant of cultural biases in their datasets while using our approach to derive subsequent techniques for explaining future behavior.
>
> 2\. **“doesn't discuss the potential harms of such technology, … This work largely ignored the possible negative societal implications of their work”**
>
> In lines 355-368, we have discussed the immediate implications of our work, which are pertinent to how our bottom-up methodological approach affects further domain-specific behavior research. We will expand this discussion to include specific applications in social robotics (line 18) and surveillance, to broaden the scope to societal implications.
>
> 3\. **Issues Acknowledged: No**
>
> We have considered and mitigated the issues surrounding personally identifiable information (faces), which is still an ethical consideration. Even though the dataset used in this work is publicly available, we have avoided displaying source videos, using 3D models in our qualitative visualizations instead. The faces in Fig. 1 from another publicly available dataset are also blurred for the same reason.

---

### Decision · Program_Chairs · 2021-09-27

**Decision:**

Reject

**Comment:**

This paper proposes a novel problem involving the prediction of nonverbal behaviors in social settings, and introduces a neural process model fit to the problem. Reviewers had no major methodological concerns, but raised concerns about novelty in the context of a machine learning methods venue, and about clarity. An extensive discussion resolved some misunderstandings, but did not yield a consensus in favor of accepting the paper to NeurIPS.